# TREND/SEASONALITY BASED CAUSAL STRUCTURE FOR TIME SERIES COUNTERFACTUAL OUTCOME PREDICTION

## ABSTRACT

In the causal effect estimation, most models have focused on estimating counterfactual outcomes in the static setting, and it is still difficult to predict the outcomes in the longitudinal setting due to time-varying confounder. To resolve the time-varying confounder issue, while the balance representation learning-based approaches have been primarily proposed, they inherently introduces a certain degree of selection bias since the balance representations act as confounders for both treatment and outcomes. In this paper, a new trend/seasonality decomposition based causal structure is proposed for the counterfactual outcome prediction in the time-series setting. We leverage a decomposition methodology to reduce the selection bias further. Specifically, it extracts meaningful decomposed representations such as confounders and adjustment variables, which help to yield more accurate treatment effect estimation with low variance. Inspired by the fact, the proposed causal structure learns trend/seasonality representations as the confounders/adjustment variables in the direction of minimizing the selection bias, and those representations are effective in the counterfactual outcome prediction especially under the long time sequence and high time-varying confounding settings. We evaluate the proposed causal structure with several trend/seasonality decomposition algorithms on synthetic and real-world datasets. From various experiments, the proposed causal structure achieves superior performance over the state-of-the-art algorithms.

## 1 INTRODUCTION

Causal effect estimation, aiming at precisely predicting the potential outcomes under 'what-if' scenarios, plays an essential role in the decision making processes. For example, the health conditions of the each patient after applying some treatments should be accurately estimated to determine the optimal timings for assigning treatment and to understand how diseases evolve under different treatment plans.

Randomized controlled trial (RCT) is the best way to learn the treatment effect, where the treatments are assigned in a random way to ensure that they are independent of the individual covariates. However, conducting RCT experiments can be expensive, and in some cases, it may be deemed to be impractical or unethical. Therefore, instead of the RCT experiments, numerous counterfactual outcome prediction methods have been proposed by using the observational data only. However, there are some challenges when predicting (time-series) counterfactual outcomes as follows:

(i) The primary challenge in counterfactual outcome prediction is that these potential outcomes are never observed. Consequently, the traditional supervised learning frameworks cannot be applied, which makes this task more difficult.

(ii) In observational dataset, there can exist few samples within each treatment, and this scarcity can lead to the poor generalization performance and inferior the performance of the counterfactual prediction at those treatments—a phenomenon termed as *selection bias*. This issue is exacerbated in the time-series domain due to the time-dependent confounders. More specifically, the input covariates can be affected by the past treatments and also concurrently impact both future treatment and outcomes, which yields a time-dependent confounding bias.

To tackle the second challenge in the time series domain, several state-of-the-art machine learning techniques have been recently introduced to predict the counterfactual outcomes over time. Recurrent neural network (RNN) based approaches have been mainly introduced, e.g., recurrent marginal structural networks (RMSNs) (Lim et al., 2018), counterfactual recurrent network (CRN) (Bica et al., 2020), and G-Net (Rui et al., 2021). Since they are based on the simple RNNs, their ability to capture complex, long-term dependencies from the observational data can be limited. To alleviate the long-term dependency problem, a transformer based architecture, called Causal Transformer (CT), was proposed (Melnychuk et al., 2022). However, most of the aforementioned algorithms focused on learning treatment invariant balance representations to reduce selection bias; therefore, they structurally induce some selection bias because the balance representations act as confounders for the treatments and outcomes (Hassanpour & Greiner, 2019).

In this paper, we propose a new trend/seasonality decomposition based causal structure for the time series counterfactual outcome prediction. For that purpose, we follow the principle of decomposition-based methods (Hassanpour & Greiner, 2019; 2020; Kuang et al., 2017), where the main idea is to identify the relevant decomposed representations from the input covariates such as the confounders and adjustment variables. Especially in Kuang et al. (2017), it should be noted that considering the adjustment variables helps to give the accurate treatment effect estimation with low variance, where the adjustment variables represent the variables affecting the outcomes only. On the other hand, the trend/seasonality representations are most meaningful underlying factors in the time series domain and effective in long sequence prediction task. Building on both ideas, if the trend/seasonality representations, acting as the confounders/adjustment variables, are learned in the direction of minimizing the selection bias, we believe that those representations can play a particularly significant role in predicting counterfactual outcomes.

Our main contributions are as follows:

- The trend/seasonality decomposition based causal structure is newly proposed for the time series counterfactual outcome prediction task. To the best of our knowledge, it is the first decomposition based causal structure under the time series setting.

- In the proposed causal structure, the trend/seasonality representations are expected to be learned in the direction of minimizing the selection bias. As a result, the performance of the counterfactual outcome prediction is improved as illustrated in the experiments.

- By the virtue of the fact that the trend/seasonality are effective in the long sequence forecasting task, the trend/seasonality representations of the proposed causal structure are also effective especially for long sequence counterfactual outcome prediction, as shown in the experiments.

## 2 RELATED WORKS

**Counterfactual outcome estimation.** In the causal effect estimation, most algorithms have predominantly focused on predicting the counterfactual outcomes from observational data in static settings, where there are no time-varying covariates, treatments, and outcomes. In the static settings, the selection bias can also be introduced because the treatments depend on the static features. To handle the selection bias under the static settings, several approaches have been proposed, such as propensity matching, balance representation, and propensity-aware hyperparameter tuning.

On the other hand, there are also several works for the counterfactual outcome prediction in the time series setting. Initial works include G-computation formula, structural nested model, marginal structural model (MSM) (Mansournia et al., 2012; Robins et al., 2000). To address more complex time-dependency scenarios, several algorithms such as Bayesian non-parametrics (Soleimani et al., 2017) or RNN (Qian et al., 2021) have been proposed to replace the part of those initial works. Recently, various fully deep learning-based approaches have been introduced with the advances in machine learning. Specifically, the recurrent marginal structural network (RMSN) (Lim et al., 2018) was proposed to improve the MSM by employing the RNN module to predict the inverse probability of treatment weight (IPTW). However, it cannot resolve the high variance problem of the propensity weights. To deal with the high variance issue in a novel way, another RNN based work, called counterfactual recurrent network (CRN) (Bica et al., 2020), was presented by adopting the idea of the representation learning to handle the time-varying confounders. More specifically, CRN extracts

a sequence of the balance representations, which are predictive to the outcomes but not predictive to the treatments for reducing the selection bias. Furthermore, G-Net (Rui et al., 2021) predicts the potential outcomes and time-varying covariates, and performs G-computation for multi-step ahead prediction. However, all the aforementioned networks were based on the RNN architecture; therefore, their performance can be restricted due to the limitation of capturing the long-range dependencies. To resolve the long-range dependency issue, causal transformer (CT) (Melnychuk et al., 2022) was also proposed with a novel counterfactual domain confusion (CDC) loss. However, most of the aforementioned networks have focused on learning the balance representation to reduce the selection bias; therefore, they cannot reduce all the selection bias since the balance representations work as a confounder for both treatment and outcomes, which inherently would introduce some bias. Consequently, their performance of the counterfactual outcome prediction can be limited.

**Reducing selection bias.** By adopting the idea of the representation learning, the pioneer work (Johansson et al., 2016) and its variants (Shalit et al., 2017) were introduced to reduce the selection bias in the causal effect estimation. The key idea is to minimize the discrepancy measures in the representation space between treated and untreated samples. However, they cannot reduce all the selection bias (Hassanpour & Greiner, 2019) since those representations work as a confounder for both treatment and outcomes. Therefore, several decomposition-based works Hassanpour & Greiner (2019); Kuang et al. (2017); Hassanpour & Greiner (2020); Wu et al. (2023) have been proposed by identifying the relevant representations and discarding the irrelevant information from the input covariates to reduce the selection bias further. Furthermore, a similarity preserved method was also proposed (Yao et al., 2018). where it attempts to maintain the same neighbourhood relationships in the learned representation space as in the original input space. In this paper, we propose a trend/seasonality decomposition-based causal structure by adopting the idea of the decomposition methods.

**Trend/Seasonality decomposition.** Time series decomposition, identifying meaningful underlying patterns in time series data, is one of the most widely used analytical methods in the field of time series analysis. Time series data is commonly assumed to be composed of three main components: trend, seasonality, and residual. Therefore, we can analyze the time series data by using the trend/seasonality decomposition techniques, which provide more insights to better understand it. Recently, many deep learning-based networks have been introduced to extract meaningful trend/seasonality representations, and those representations have been shown to yield a superior performance in the various downstream tasks. For example, in the time series forecasting, the trend representations are extracted by using the average pool or convolution neural network (Wu et al., 2021; Woo et al., 2022; Zeng et al., 2023); on the other hand, the seasonality representations are learned by using the discrete Fourier transform in the frequency domain (Zhou et al., 2022; Woo et al., 2022). Furthermore, those trend/seasonality representations can be also extracted in a multiresolution way (Wang et al., 2023). In the prediction task, the trend/seasonality based algorithms extract the trend/seasonality representations predictive to the future outcomes and they have been shown a superior performance. In this paper, the trend/seasonality representations are learned in the direction of minimizing the selection bias for the counterfactual outcome prediction task.

## 3 PROBLEM SETTINGS

Consider the following time-varying observational dataset $D = \{\mathbf{x}_t^{(i)}, \mathbf{a}_t^{(i)}, \mathbf{y}_{t+1}^{(i)}\}$ with the static covariates $\mathbf{v}^{(i)}$. For each $i^{th}$ time series data, $\mathbf{x}_t^{(i)}$ denotes input covariates at time $t$, $\mathbf{a}_t^{(i)} \in \{a_1, \dots, a_d\}$ is defined by $d$ categorical treatments, and $\mathbf{y}_{t+1}^{(i)}$ represents outcomes at time $t + 1$. For the sake of the simplicity, the superscript $(i)$ will be omitted from now on unless it is necessary. We first define following two arbitrary vectors, $\overleftarrow{\mathbf{Q}}_t$ with the past $P$ time steps and $\overrightarrow{\mathbf{Q}}_t$ with the future $F$ time steps, respectively:

$$\overleftarrow{\mathbf{Q}}_t = [\mathbf{q}_{t-P+1}, \dots, \mathbf{q}_t] \tag{1}$$

$$\overrightarrow{\mathbf{Q}}_t = [\mathbf{q}_t, \dots, \mathbf{q}_{t+F-1}]. \tag{2}$$

In addition, the sequence of an arbitrary vector $\bar{\mathbf{Q}}_t$ until time stamp $t$ is denoted by

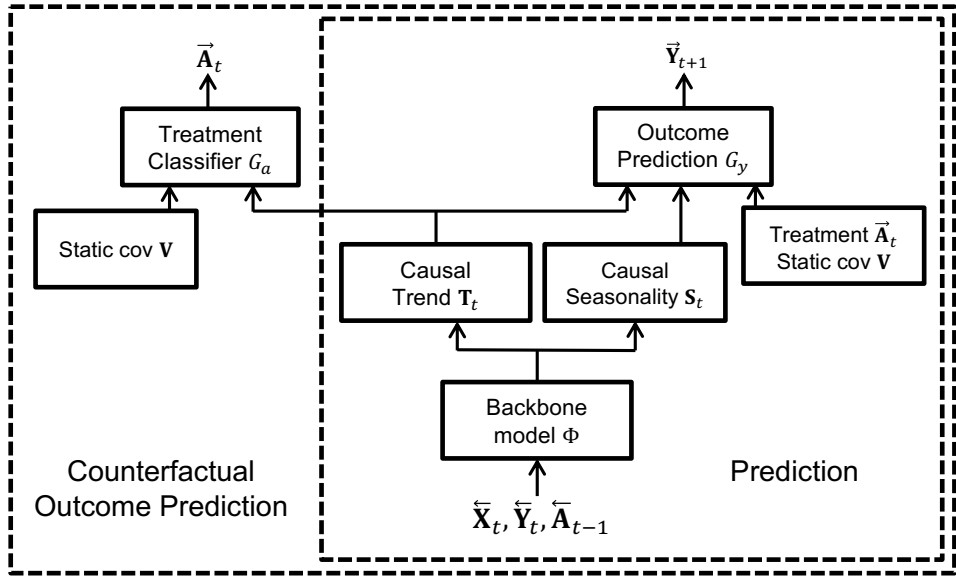

Figure 1: The proposed causal structure for the counterfactual outcome prediction.

$$\bar{\mathbf{Q}}_t = (\mathbf{q}_1, \ldots, \mathbf{q}_t). \tag{3}$$

Then, given $\overleftrightarrow{\mathbf{H}}_t = \{\overleftarrow{\mathbf{X}}_t, \overleftarrow{\mathbf{A}}_{t-1}, \overleftarrow{\mathbf{Y}}_t, \mathbf{V}\}$, we want to predict the next $F$ time steps of the potential outcomes $\overrightarrow{\mathbf{Y}}_{t+1}$ under all possible sequences of treatments $\overrightarrow{\mathbf{A}}_t$, i.e., $E\{\overrightarrow{\mathbf{Y}}_{t+1}\left(\overrightarrow{\mathbf{A}}_t\right)|\overleftarrow{\mathbf{H}}_t\}$. Here, the previous input covariates $\overleftarrow{\mathbf{X}}_t$, previous outcomes $\overleftarrow{\mathbf{Y}}_t$, and previous treatments $\overleftarrow{\mathbf{A}}_{t-1}$ are defined by using equations (1)-(2) as follows:

$$\overleftarrow{\mathbf{X}}_t = [\mathbf{x}_{t-P+1}, \ldots, \mathbf{x}_t] \tag{4}$$

$$\overleftarrow{\mathbf{Y}}_t = [\mathbf{y}_{t-P+1}, \ldots, \mathbf{y}_t] \tag{5}$$

$$\overleftarrow{\mathbf{A}}_{t-1} = [\mathbf{a}_{t-P}, \ldots, \mathbf{a}_{t-1}]. \tag{6}$$

Furthermore, three standard assumptions are supposed to hold for identifying the treatment effect as follows:

**Assumption 1: Consistency.** The potential outcome under the realized treatment sequence $\bar{\mathbf{a}}_t$ is the same as the factual outcome, i.e., $\mathbf{Y}_{t+1}\left(\bar{\mathbf{a}}_t\right) = \mathbf{Y}_{t+1}$.

**Assumption 2: Positivity.** Let $\bar{\mathbf{H}}_t = \{\bar{\mathbf{X}}_t, \bar{\mathbf{A}}_{t-1}, \bar{\mathbf{Y}}_t, \mathbf{V}\}$. If $P(\bar{\mathbf{H}}_t = \bar{\mathbf{h}}_t) > 0$, then $P(\mathbf{A}_t = \mathbf{a}_t|\bar{\mathbf{H}}_t = \bar{\mathbf{h}}_t) > 0$ for all $\bar{\mathbf{A}}_t$, where $\bar{\mathbf{h}}_t$ is some realization of $\bar{\mathbf{H}}_t$. It means that the probability of any treatment is non-zero for all the history space over time.

**Assumption 3: Unconfoundedness.** Given the observed history $\bar{\mathbf{H}}_t$, the current treatment $\mathbf{A}_t$ is independent of the potential outcome. Formally, $\mathbf{Y}_{t+1}\left(\mathbf{a}_t\right) \perp\!\!\!\perp \mathbf{A}_t|\bar{\mathbf{H}}_t, \forall \mathbf{a}_t$. It means that there are no unobserved confounders. That is, all confounders affecting both treatments and outcomes are given in the observed dataset.

## 4 CAUSAL TREND/SEASONALITY BASED CAUSAL STRUCTURE

### 4.1 PROPOSED CAUSAL STRUCTURE

We introduce a new trend/seasonality decomposition based causal structure for the counterfactual outcome estimation in the time series setting, as shown in Figure 1. As discussed in Introduction,

extracting decomposed representations such as confounders and adjustment variables can lead to the accurate treatment effect estimation with low variance (Kuang et al., 2017). Based on this idea, while the trend representations are assumed to work as confounders affecting both outcomes and treatments, the seasonality representations are assumed to work as adjustment variables influencing the outcomes only, as depicted in Figure 1. This assumption is reasonable since the treatment variables are more closely related with the trend representations rather than the small scale of the seasonality representations. For example, when we decide to promote some items in the sales forecasting, we usually consider their demand trend rather than the small scale of seasonality.

As illustrated in Figure 1, the trend representation $\mathbf{T}_t$ and seasonality representation $\mathbf{S}_t$ are extracted through the backbone decomposition models.

$$\mathbf{T}_t, \ \mathbf{S}_t = \Phi\left(\overleftarrow{\mathbf{X}}_t, \overleftarrow{\mathbf{A}}_{t-1}, \overleftarrow{\mathbf{Y}}_t; \theta_r\right) \tag{7}$$

After that, the trend/seasonality representations are concatenated with the static and treatment variables to estimate both treatments and potential outcomes using fully-connected linear layers $G_y$ for the outcome prediction and $G_a$ for the treatment classifier as follows:

$$\overrightarrow{\mathbf{Y}}_{t+1} = G_y(\mathbf{T}_t, \mathbf{S}_t, \overrightarrow{\mathbf{A}}_t, \mathbf{V}; \theta_y) \tag{8}$$

$$\overrightarrow{\mathbf{A}}_t = G_a(\mathbf{T}_t, \mathbf{V}; \theta_a). \tag{9}$$

In the following, we discuss the characteristics of the proposed causal structure in comparisons with the trend/seasonality decomposition based time series forecasting algorithms.

- The trend/seasonality representations are learned differently between the time series forecasting algorithms and proposed causal structure. The trend/seasonality representations for time series forecasting algorithms are supposed to be predictive to the future outcomes only (see prediction of Figure 1). On the other hand, in the counterfactual outcome prediction, the trend/seasonality representations for the proposed causal structure should be predictive to both future outcomes and treatments (see counterfactual outcome prediction of Figure 1). Therefore, we call those representations as causal trend/seasonality representations here. The performance difference between the time series forecasting algorithms and proposed causal structure will be described in the ablation study of the experiments.

- It is well known that the trend/seasonality representations are effective in the long time series forecasting task. In the proposed causal structure, the causal trend/seasonality representations are trained in the direction of minimizing the selection bias as well; therefore, the causal representations are expected to be effective in the long sequence counterfactual outcome prediction, which will be shown in the experimental results.

## 4.2 Loss

To improve the performance of the counterfactual outcome prediction, the proposed causal structure should be trained in the direction of minimizing the selection bias. To this end, the widely used loss functions in the decomposition methods are naturally extended to the time series case here. Specifically, the total loss function consists of factual loss, imbalance loss, and cross entropy loss. Each loss function will be explained in more detail.

**Factual loss.** After predicting the potential outcomes as in equation (8), the estimated potential outcomes should be close to the true outcomes. For that purpose, a simple mean squared error (MSE) function was used for the factual loss $\mathcal{L}_F$ as

$$\mathcal{L}_F = \left\|\overrightarrow{\mathbf{Y}}_{t+1} - G_y\left(\mathbf{T}_t, \mathbf{S}_t, \overrightarrow{\mathbf{A}}_t, \mathbf{V}; \theta_y\right)\right\|^2. \tag{10}$$

Note that while the simple MSE function was employed here, its weighted version (Hassanpour & Greiner, 2020) can be utilized instead.

**Imbalance loss.** As described in Figure 1, the selection bias of the proposed causal structure inherently is introduced by the causal trend representations. In this case, the selection bias is minimized by making the causal seasonality representations be independent of the treatments. To achieve the goal, the following discrepancy function should be minimized:

$$\text{disc}\left(\{\mathbf{S}_t\}_{\mathbf{a}_t:=a_i}, \{\mathbf{S}_t\}_{\mathbf{a}_t:=a_j}\right), \quad a_i \neq a_j. \tag{11}$$

For multiple treatments scenario, we use the following for the imbalance loss $\mathcal{L}_I$:

$$\mathcal{L}_I = \sum_{a_i \neq a_j} \text{disc}\left(\{\mathbf{S}_t\}_{\mathbf{a}_t:=a_i}, \{\mathbf{S}_t\}_{\mathbf{a}_t:=a_j}\right) \tag{12}$$

In equations (11)-(12), the simple maximum mean discrepancy (MMD) function is used to compute the dissimilarity between two conditional distributions. Other discrepancy functions are also acceptable, e.g., Wasserstein distance. By minimizing the imbalance loss, the learned causal seasonality representations have no information about the treatments, and all the necessary confounding factors are expected to be embedded in the causal trend representations. Therefore, the selection bias of the proposed causal structure is effectively reduced by minimizing the imbalance loss.

**Cross Entropy loss.** As discussed above, we assume that the causal trend representations have enough information to choose the treatments by minimizing the imbalance loss. Therefore, we can predict the next treatments by minimizing the cross entropy loss as follows:

$$\mathcal{L}_C = -\sum_{i,t} \log \overrightarrow{\mathbf{A}}_{i,t}\left(G_a\left(\mathbf{T}_t, \mathbf{V}; \theta_a\right)_i\right). \tag{13}$$

Finally, the total loss function of the proposed causal structure is given as follows:

$$\mathcal{L} = \mathcal{L}_F + \lambda_1 \mathcal{L}_I + \lambda_2 \mathcal{L}_C. \tag{14}$$

The proposed causal structure is trained using equation (14) in an end-to-end fashion.

## 5  EXPERIMENTAL RESULTS

As discussed before, the counterfactual outcomes are not known for the real-world data. Therefore, the counterfactual prediction performance of the proposed causal structure should be assessed on the synthetic and semi-synthetic dataset. For that purpose, the proposed causal structure will be tested on i) the synthetic dataset based on tumor growth model and ii) the semi-synthetic dataset created by the widely used real MIMIC-III dataset in the counterfactual inference under the time series setting. Finally, we will also evaluate the factual prediction performance on the real MIMIC-III dataset. Note that for the synthetic and semi-synthetic dataset, while only observational data is given in the train and validation sets, all possible counterfactual outcomes can be generated with the respect to all possible treatments for each time-stamp to test the proposed causal structure. For a fair comparison, we performed the experiments under identical environments as in Melnychuk et al. (2022); therefore, detailed implementations of the data generation processes are referred to Melnychuk et al. (2022). For all experiments, $\lambda_1 = \lambda_2 = 1$ was used in the loss function, and the past time stamp $P = 15$ and the future time stamp $F = 15$ were set to test the proposed causal structure under the long sequence scenario. All performance was evaluated by averaging the results over three runs with different random seeds.

### 5.1  MODEL OF TUMOR GROWTH

For the synthetic data experiments, we used the tumor growth model. More specifically, the volume of tumour $V_t$ at time stamp $t$ was modelled as follows:

$$V_{t+1} = \left(1 + \rho \log\left(\frac{K}{V_t}\right) - \beta_c C_t - \left(\alpha_\gamma d_t + \beta_\gamma d_t^2 t\right) + e_t\right) V_t, \tag{15}$$

where $C_t$ and $d_t$ are denoted by chemotherapy concentration and radiotherapy dose, respectively, $e_t \sim N(0, 0.01^2)$ is a noise term. Other variables (i.e., $\rho, K, \beta_c, \alpha_\gamma, \beta_\gamma$) are simulation parameters. In this model, the time-varying confounding factor is modelled by assigning chemotherapy and radiotherapy terms as Bernoulli random variables with probabilities $P_t^c$ and $P_t^r$ as follows:

$$P_t^c = \sigma \left( \frac{\gamma_c}{D_{\max}} \left( \tilde{D}_t - \delta_c \right) \right) \tag{16}$$

$$P_t^r = \sigma \left( \frac{\gamma_r}{D_{\max}} \left( \tilde{D}_t - \delta_r \right) \right), \tag{17}$$

where $\tilde{D}_t$ is the averaged diameter of the tumor over last 15 days, $D_{\max} = 13$cm, $\sigma(\cdot)$ is the sigmoid function, and $\delta_c = \delta_r = D_{\max}/2$. When $\gamma_c$ and $\gamma_r$ are higher, the amount of the time-varying confounding is higher and vice versa. In the tumor growth model, the outcome is the volume of tumor $V_t$, and four treatments were given: (i) no treatment, (ii) radiotherapy, (iii) chemotherapy, (iv) both radiotherapy and chemotherapy. We generated 10,000 patient trajectories for training set, 1,000 for validation set, and 1,000 for test set. The maximum length of the generated sequences is limited to 60 time steps.

## 5.2 MIMIC BASED SEMI-SYNTHETIC AND REAL DATASET

We also tested the proposed causal structure on the semi-synthetic MIMIC-III dataset (Johnson et al., 2016). We used 25 vital signs as time-varying covariates and 3 static covariates (i.e., gender, ethnicity, and age). In addition, diastolic blood pressure variable was used as the outcome variable, and vasopressors and mechanical ventilation were considered as the treatments. We generated the counterfactual outcomes under endogenous and exogenous setting as in Schulam & Saria (2017).

## 5.3 BASELINE MODELS

To evaluate the performance of the proposed causal structure, the following state-of-the-art models were used: Counterfactual Recurrent Network (CRN) (Bica et al., 2020) and Causal Transformer (CT) (Melnychuk et al., 2022). In the proposed causal structure, the recent trend/seasonality decomposition based time series forecasting models can be used to extract the causal trend/seasonality representations. For that purpose, three recent decomposition models such as FEDformer (Zhou et al., 2022), DLinear (Zeng et al., 2023), and MICN (Wang et al., 2023), were employed for the proposed causal structure. According to each decomposition model, we named three models as Causal FEDformer, Causal DLinear, Causal MICN, respectively.

## 5.4 PERFORMANCE COMPARISON

The performance was evaluated in terms of root mean squared error (RMSE) for all datasets.

**Performance on synthetic dataset.** In Figure 2, the performance of the synthetic dataset was given along the time stamps for various time-varying confounders $\gamma = \gamma_c = \gamma_r$, and its time-averaged results were also listed in Table 1. To test the proposed causal structure under more realistic setting, a random trajectory setting in CT (Melnychuk et al., 2022) was considered here, where the random trajectory setting is to assign the treatments in a randomized way. As illustrated in Table 1, the proposed causal structure generally achieved better performance than the baseline models on average. Especially for Causal DLinear and Causal MICN, the performance gain over the baselines is higher for the longer time stamp $\tau$ and larger $\gamma$ as shown in (d) of Figure 2.

**Performance on semi-synthetic and real dataset.** In Table 2, the performance of the semi-synthetic and real dataset was given. For the semi-synthetic dataset, the proposed causal structure based algorithms showed better results on average when compared with the baselines. Furthermore, similar to the synthetic results, the proposed causal structure obtained a larger performance gain in case of longer time steps $\tau$. For the real dataset, while the three proposed causal structure based models have a similar result, they achieved a better performance than the baselines.

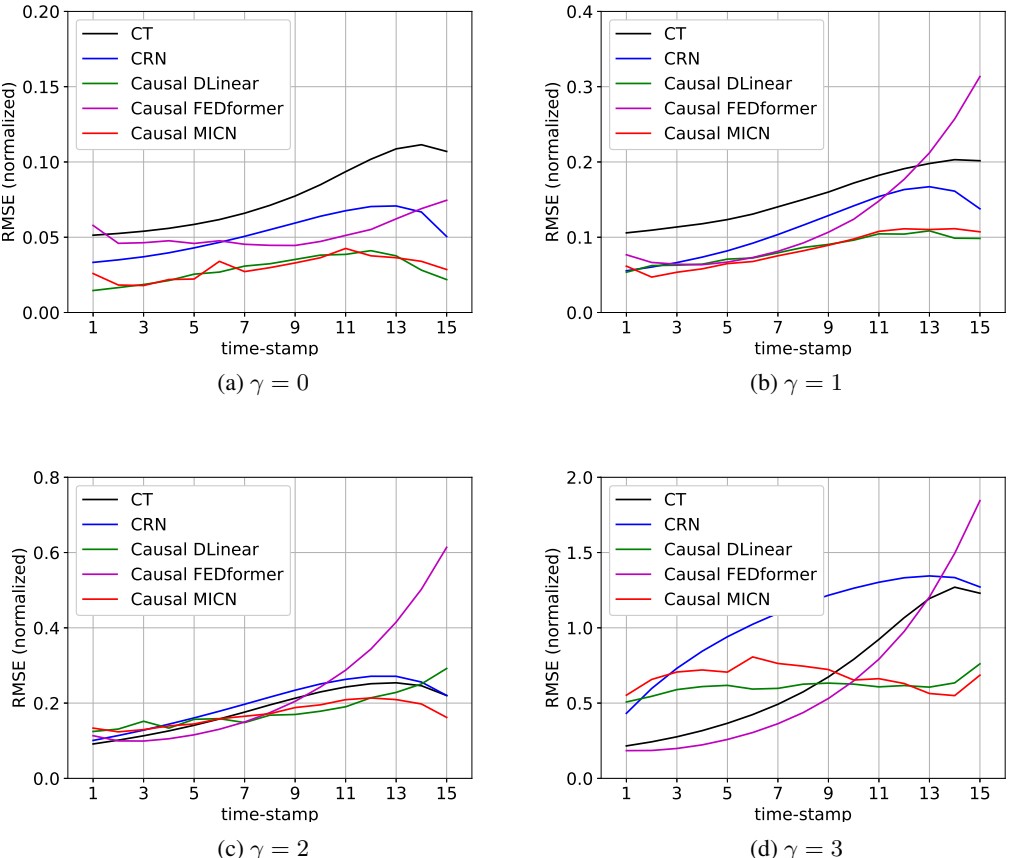

Figure 2: The performance of the counterfactual outcome prediction on the synthetic dataset.

Table 1: The time-averaged performance of the counterfactual outcome prediction on the synthetic dataset.

|  | $\gamma = 0$ | $\gamma = 1$ | $\gamma = 2$ | $\gamma = 3$ |
|---|---|---|---|---|
| CRN | 0.052 | 0.113 | 0.200 | 1.059 |
| CT | 0.077 | 0.153 | 0.183 | 0.670 |
| Causal DLinear | **0.028** | 0.083 | 0.179 | **0.611** |
| Causal FEDformer | 0.052 | 0.128 | 0.239 | 0.642 |
| Causal MICN | 0.029 | **0.082** | **0.169** | 0.674 |

Table 2: The performance of the semi-synthetic and real dataset.

| | CRN | | CT | | Causal DLinear | | Causal FEDformer | | Causal MICN | |
|---|---|---|---|---|---|---|---|---|---|---|
| | semi | real | semi | real | semi | real | semi | real | semi | real |
| $\tau = 1$ | 0.566 | 9.210 | 0.597 | 9.068 | 0.500 | 6.084 | **0.414** | 5.926 | 0.597 | **5.422** |
| $\tau = 2$ | 0.718 | 9.819 | 0.730 | 9.607 | 0.623 | 8.792 | **0.564** | 8.836 | 0.714 | 8.867 |
| $\tau = 3$ | 0.718 | 10.110 | 0.813 | 9.948 | 0.725 | 9.301 | **0.647** | 9.296 | 0.778 | 9.359 |
| $\tau = 4$ | 0.908 | 10.355 | 0.867 | 9.682 | 0.790 | 10.223 | **0.699** | 9.649 | 0.831 | 9.722 |
| $\tau = 5$ | 0.974 | 10.554 | 0.904 | 10.456 | 0.832 | 9.950 | **0.756** | 9.921 | 0.876 | 10.003 |
| $\tau = 6$ | 1.033 | 10.817 | 0.930 | 10.671 | 0.869 | 10.183 | **0.798** | **10.150** | 0.891 | 10.300 |
| $\tau = 7$ | 1.085 | 11.064 | 0.947 | 10.857 | 0.888 | 10.347 | **0.827** | **10.334** | 0.917 | 10.485 |
| $\tau = 8$ | 1.087 | 11.311 | 0.958 | 10.957 | 0.909 | 10.528 | **0.823** | **10.514** | 0.909 | 10.657 |
| $\tau = 9$ | 1.088 | 11.553 | 0.968 | 11.103 | 0.930 | 10.706 | **0.816** | **10.685** | 0.906 | 10.818 |
| $\tau = 10$ | 1.084 | 11.753 | 0.976 | 11.215 | 0.941 | 10.823 | **0.826** | **10.822** | 0.905 | 10.957 |
| $\tau = 11$ | 1.081 | 11.970 | 0.982 | 11.283 | 0.959 | **10.938** | **0.825** | 10.939 | 0.914 | 11.079 |
| $\tau = 12$ | 1.074 | 12.145 | 0.986 | 11.435 | 0.972 | 11.072 | **0.818** | **11.066** | 0.911 | 11.215 |
| $\tau = 13$ | 1.070 | 12.314 | 0.988 | 11.562 | 0.989 | 11.197 | **0.820** | **11.195** | 0.905 | 11.346 |
| $\tau = 14$ | 1.065 | 12.436 | 0.991 | 11.672 | 0.997 | 11.284 | **0.824** | **11.255** | 0.904 | 11.421 |
| $\tau = 15$ | 1.062 | 12.559 | 0.993 | 11.845 | 1.005 | 11.417 | **0.815** | **11.391** | 0.898 | 11.650 |
| Average | 0.974 | 11.198 | 0.909 | 10.790 | 0.862 | 10.153 | **0.751** | **10.132** | 0.857 | 10.220 |

Table 3: Ablation study for the proposed causal structure. Note that for $\lambda_1 = \lambda_2 = 0$, the proposed causal structure was trained using the factual loss only.

| | $\gamma = 0$ | $\gamma = 1$ | $\gamma = 2$ | $\gamma = 3$ |
|---|---|---|---|---|
| Causal DLinear ($\lambda_1 = \lambda_2 = 0$) | **0.025** | 0.086 | **0.179** | 0.686 |
| Causal DLinear | 0.028 | **0.083** | **0.179** | **0.611** |

## 5.5 ABLATION STUDY

We performed an ablation study of the proposed causal structure to confirm its effectiveness. For that purpose, we set $\lambda_1 = \lambda_2 = 0$ in the loss function to train the proposed causal structure using the factual loss only, which implies that the learned trend/seasonality representations are predictive to the outcomes only as in the prediction task. By doing so, the effect of minimizing the selection bias will be investigated. In Table 3, the time-averaged results were listed for Causal DLinear ($\lambda_1 = \lambda_2 = 0$) and Causal DLinear trained with the total loss function. Note that the performance of Table 3 was evaluated on the synthetic dataset. From the results, we can see that both results were similar in case of low time-varying confounding (i.e., $\gamma = 0$–2), and Causal DLinear showed the better results than Causal DLinear with $\lambda_1 = \lambda_2 = 0$ when the time-varying confounding is large (i.e., $\gamma = 3$). It means that the imbalance loss effectively minimizes the selection bias and finally the performance of the counterfactual outcome prediction is improved.

## 6 CONCLUSION AND FUTURE WORKS

In this paper, a new trend/seasonality decomposition based causal structure was proposed for the time series counterfactual outcome prediction. From the various experiments, we showed that the causal trend/seasonality representations can effectively reduce the selection bias in the long time sequence and high time-varying confounding settings. We believe that our proposed perspective can provide new insight and direction on the time series counterfactual outcome prediction. We think that it will be a good research topic to investigate novel decomposed representations for the time series counterfactual prediction.

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
