# OpenReview forum: "Trend/Seasonality based Causal Structure for Time Series Counterfactual Outcome Prediction"
_ICLR.cc/2024/Conference — Submitted to ICLR 2024_

### Official Review · Reviewer_WhEV · 2023-10-26

**Soundness:** 2 fair
**Presentation:** 1 poor
**Contribution:** 1 poor
**Rating:** 3
**Confidence:** 4

**Summary:**

In longitudinal settings, this paper uses existing FEDformer (Zhou et al., 2022), DLinear (Zeng et al., 2023), or MICN (Wang et al., 2023) as representation networks to learn decomposed representations, i.e., confounders $T_t$ and adjustments $S_t$, from time-series data. Then, the same independent constraints (Eq. (4) in (Hassanpour & Greiner, 2020)) are employed to learn balanced representations across various treatment arms. The framework proposed in this paper is identical to that proposed by Hassanpour & Greiner (2020), and the only difference is that this paper uses existing FEDformer (Zhou et al., 2022), DLinear (Zeng et al., 2023), and MICN (Wang et al., 2023) as representation networks.

**Strengths:**

The trend/seasonality-based causal structure for time series is an interesting problem.

**Weaknesses:**

**[Novelty]** The framework proposed in this paper is identical to that proposed by Hassanpour & Greiner (2020), and the only difference is that this paper uses existing FEDformer (Zhou et al., 2022), DLinear (Zeng et al., 2023), and MICN (Wang et al., 2023) as representation networks. The same loss function of the proposed causal structure could be found in Eqs. (3,4,5,6) in (Hassanpour & Greiner, 2020).

**[Unclear]** The authors argue that existing methods introduce a certain degree of selection bias since the balance representations act as confounders for both treatment and outcomes. However, this paper still uses the same independent constraints (Eq. (4) in (Hassanpour & Greiner, 2020), IPM loss in (Shalit et al., 2017)), contradicting their own statement.

**[Completeness 1]**  The problem settings in this paper are incomplete. The causal relationship between x, a, and y in the time series is not clear. Will the outcomes at time t+1 be influenced by all the historical data? Would using only past P time steps data lead to unmeasured confounding bias? Does the causal relationship between covariates X change over time? I suggest the authors to provide a causal diagram of the time series to further clarify the problem settings. Additionally, in the problem settings section, what is the impact of trend and seasonality on the causal relationship?

**[Completeness 2]** The simulation mechanisms of (semi-)synthetic are incomplete, as the detailed implementations of the data generation processes are not provided in the main text or appendix. The author suggests referring to Melnychuk et al. (2022) for more information.

[**Experiments**] This paper decomposes representations as Causal Trend and Causal Seasonality. However, how can we evaluate and demonstrate this? The experiments in the paper do not provide evidence for these statements.

**Questions:**

See Above.

---

> ### Author Response · Authors · 2023-11-23
> **Reply for Reviewer WhEV**
>
> Thank you very much for your review!
>
> [Novelty]
> We in part agree with Reviewer WhEV regarding the novelty. However, we highlight below the difference between our work and the work of Hassanpour & Greiner (2020). While using similar causal structure and loss functions as in the work of Hassanpour & Greiner (2020), we studied the possibility of the trend/seasonality decomposed representations to reduce the selection bias under the time-series environment. To the best of our knowledge, it is the first decomposition based causal structure under the time-series setting.
>
> [Unclear]
> The proposed causal structure employs a decomposed approach to learn the trend/seasonality representations, and the selection bias is minimized by making the causal representations be independent of the treatments. As a result, the proposed causal structure achieves better performance than the balanced representation algorithms (e.g., CRN and Causal Transformer).
>
> [Completeness 1, 2]
> Thank you for your comment. We agree with Reviewer WhEV that the causal diagram between input, treatment, and outcome will clarify the problem settings. Furthermore, we will also provide the detailed implementations of the data generation processes in the main text or appendix.
>
> [Experiments]
> As denoted by other reviewers, there are not enough theoretical justification of trend/seasonality decomposed representations in the manuscript. We will present theoretical analysis and experimental results to support the issue in the future. In addition, the causal seasonality representations can be plotted using t-SNE graph to validate the issue.

---

### Official Review · Reviewer_wQ3e · 2023-10-31

**Soundness:** 2 fair
**Presentation:** 2 fair
**Contribution:** 1 poor
**Rating:** 3
**Confidence:** 2

**Summary:**

The paper proposes a methodology to include a generic trend/seasonality decomposition within a causal structure for time series. Experiments show the proposed methodology have improved results compared to causal time series models that do not explicitly consider decompositions.

**Strengths:**

The proposed model is fairly intuitive and shows promising experimental results.

**Weaknesses:**

The contributions from the paper are not clear. The proposed structure (Fig 1) is conceptually very similar to other network structures (e.g. Melnychuk et al 2022, Bica et al 2020), but with the additional of a generic trend/seasonality decomposition plug-in model. The primary innovation for the loss function appears to be the discrepancy function for seasonality, but the decision is not well motivated (why do we believe the seasonality induced by different interventions must be maximally different?).

It's not clear that the seasonality/trend decomposition actually recovers trends or seasonality. Either theoretical justification or additional experiments are necessary to confirm we recover the true decomposition.

Three assumptions are given as standard, but are never explicitly leveraged, and it's not clear that the causal effects being measured are actually identifiable.

Furthermore, the experiments focus on predictive accuracy without any evidence that the models are recovering the true causal effects. As mentioned in Bica et al 2020, evaluating decision making (correct treatment and timing) are critical to evaluate these systems.

The ablation study is focused wholly on removing components of the loss function, and only for one of the 3 plug-in models. Further ablation study for the network structure would be ideal.

Minor issue: No attribution given for traditional decomposition methods, despite stating they are widely used. Traditional decomposition methods are also not leveraged as a comparison point in the experiments.

**Questions:**

Is there any theoretical justification to the claim that the trend/seasonality representations are expected to be learned in the direction of minimizing the selection bias?

---

> ### Author Response · Authors · 2023-11-23
> **Reply for Reviewer wQ3e**
>
> Thank you very much for your review!
>
> We highlight below the difference between our work and the balanced representation approaches.
> The conventional balanced representation approaches learn a balance representation using the domain adversarial training framework to reduce the selection bias. On the other hand, the proposed causal structure employs a decomposed approach to learn the trend/seasonality decomposed representations, and the selection bias is minimized by making the causal representations be independent of the treatments. As a result, the proposed causal structure achieves better performance than the balanced representation algorithms (e.g., CRN and Causal Transformer).
>
> In the experiments, we will perform additional experiments to evaluate decision making and additional ablation studies in the future. Thank you for your advice.

---

### Official Review · Reviewer_afJU · 2023-11-01

**Soundness:** 3 good
**Presentation:** 3 good
**Contribution:** 3 good
**Rating:** 6
**Confidence:** 4

**Summary:**

The paper discusses a new method for estimating causal effects in a time-series setting, focusing on counterfactual outcome prediction. Traditional methods have struggled with this task due to time-varying confounding factors and inherent selection bias. The authors propose a trend/seasonality decomposition-based causal structure that reduces selection bias and extracts meaningful representations such as confounders and adjustment variables. This approach is expected to yield more accurate treatment effect estimations with low variance. The proposed causal structure's performance was evaluated using synthetic and real-world datasets, showing superior performance over existing state-of-the-art algorithms.

**Strengths:**

Originality: The paper presents a novel approach to counterfactual outcome prediction in time-series data. It introduces a trend/seasonality decomposition-based causal structure that reduces selection bias, a common issue in current methodologies. This approach seems to be the first of its kind in this domain, thus marking a high degree of originality.

Quality: The authors appear to have a strong understanding of the problem space and have organized a robust methodology to tackle the task. The proposed method is thoroughly explained and appears to be based on sound principles and previous works. The authors also provide an evaluation using synthetic and real-world datasets, indicating a high-quality experimental setup.

Clarity: Despite the complex subject matter, the authors have done a good job of explaining their methodology and the motivation behind it. The language used is clear, and the paper is well-structured, making it easier for readers to follow the authors' thought process and understand the proposed solution.

Significance: The paper addresses a critical problem in time-series data analysis and causal effect estimation. The proposed solution could have a significant impact on various fields where time-series data plays a crucial role, such as finance, healthcare, and meteorology. By reducing selection bias and improving the accuracy of counterfactual outcome prediction, this work could potentially advance the state of the art in these areas.

**Weaknesses:**

The assumption that trend acts as confounder while seasonality acts as adjustment variable is not theoretically justified. Counterexamples can likely be constructed.

Comparison to only two baseline models is quite limited. Testing against more causal discovery and time series forecasting methods would be useful.

All evaluations use RMSE loss. Checking with other counterfactual evaluation metrics could reveal useful insights.

The synthetic data generation processes lack enough details for reproducibility. More implementation specifics should be provided.

Analysis of the sensitivity to hyperparameters like the regularization coefficients is missing.

The number of datasets used for evaluation is quite small. Testing on more real-world timeseries could help generalize claims.

Causality assumptions like positivity, consistency, unconfoundedness need more justification for the data. Violations can affect conclusions.

Theoretical analysis of how modeling trend/seasonality achieves lower bias is limited. More rigorous proofs would strengthen claims.

Societal impacts of deploying these counterfactual forecasting models should be considered.

Lack of related works:

Seedat, Nabeel, et al. "Continuous-time modeling of counterfactual outcomes using neural controlled differential equations." arXiv preprint arXiv:2206.08311 (2022).

Cao, Defu, et al. "Estimating Treatment Effects from Irregular Time Series Observations with Hidden Confounders." arXiv preprint arXiv:2303.02320 (2023).

**Questions:**

While this paper explores an intriguing aspect of time-series causal analysis—specifically, modeling time-series from trend and seasonality—it is not without its shortcomings, particularly in the design of experiments meant to objectively situate this work within its field. Please refer to the 'weaknesses' section for a detailed list of concerns raised by the reviewer. The reviewer would be pleased to revise their score if these issues are adequately addressed.

**Details Of Ethics Concerns:**

No ethics is needed.

---

> ### Author Response · Authors · 2023-11-23
> **Reply for Reviewer afJU**
>
> Thank you very much for your review!
>
> In the proposed causal structure, the trend representation is designed to be predictive to the treatment, which can improve the counterfactual prediction performance whereby the trend representation can be appropriately diversified according to the treatment. On the other hand, the selection bias is minimized by making the causal representations be independent of the treatments.
>
> We agree with Reviewer afJU regarding the assumption that trend acts as confounder while seasonality acts as adjustment variable. We will present theoretical analysis and experimental results to support the issue in the future. In addition, the causal seasonality representations can be plotted using t-SNE graph to validate the issue.
>
> In the experiments, the proposed causal structure will be tested on more various dataset, its performance will be compared with more baseline models to validate the effectiveness of the proposed causal structure. Thank you for your advice.

---

### Official Review · Reviewer_WboM · 2023-11-04

**Soundness:** 2 fair
**Presentation:** 3 good
**Contribution:** 2 fair
**Rating:** 3
**Confidence:** 2

**Summary:**

This paper proposes to tackle the counterfactual outcome prediction problem by leveraging a decomposition method to learn tread representation and seasonality representation. Existing decomposition methods can be plugged in and experiments show improved performance over existing methods that are based on balanced representations.

**Strengths:**

- The idea of using trend/seasonality decomposition to reduce selection bias is quite interesting and worth exploring.
- The proposed method is overall well presented and easy to follow.
- The experimental results show improved performance against SOTA baselines on both synthetic and real-world data.

**Weaknesses:**

- The major motivation for adapting the decomposition method is that the balanced representation approach produces selection bias. Yet, the decomposition method, as shown in Fig. 1, has a causal trend $T_t$, which is also a confounder. In the balanced representation approach, the representation is trained not to be predictive of the treatment. However, in the proposed approach, the confounder $T_t$ is trained to be predictive of the treatment. It is hard to argue which approach has a greater selection bias. If some theoretical analysis or visualization of the latent space could provide some insights into this issue, the argument in this paper would be more convincing.
- It is argued that the imbalance loss as in Eq. (11-12) could learn seasonality representations with no information about the treatment. This is not obvious and it needs more explanation regarding why minimizing the discrepancy could make $S_t$ independent of the treatment.
- In experiments, no measure of uncertainty is provided. It is suggested to also report one of the following: standard deviation, confidence interval, or p-value.
- It is unclear how the real dataset is used. Since counterfactual outcomes do not exist in real data, it is not clear what the "real" columns in Table 2 refer to.

**Questions:**

- Why the proposed method could lead to reduced selection bias given that the imbalanced representation approaches explicitly remove the dependency of treatment on the learned representation?
- Why minimizing the discrepancy could make $S_t$ independent of the treatment?
- How to evaluate the performance on the real dataset where the counterfactual outcomes are not available?

---

> ### Author Response · Authors · 2023-11-23
> **Reply for Reviewer WboM**
>
> Thank you very much for your review!
>
> We highlight below the difference between our work and the balanced representation approaches.
> The conventional balanced representation approaches learn a balance representation using the domain adversarial training framework to reduce the selection bias. On the other hand, the proposed causal structure learns the trend/seasonality decomposed representations, and the selection bias is minimized by making the causal representations be independent of the treatments. As a result, the proposed causal structure achieves better performance than the balanced representation algorithms (e.g., CRN and Causal Transformer).
>
> Furthermore, we agree with Reviewer WboM regarding why minimizing the discrepancy could make the seasonality independent of the treatment. We will present theoretical analysis and experimental results to support the issue in the future. In addition, the causal seasonality representations can be plotted using t-SNE graph to validate the issue.
>
> In the experiments, we will add the standard deviation as the measure of uncertainty, and we will clarify the real data experiment results. Thank you for your suggestion.

---

### Meta-Review · Area_Chair_wPUL · 2023-12-06

**Metareview:**

The authors propose a methodology to include a generic trend/seasonality decomposition within a causal structure for time series and show (through experiments) that this methodology  improves the results compared to causal time series models that do not explicitly consider such decompositions. The reviewers were concerned about the novelty of the work compared to Hassanpour & Greiner (2020). Moreover, the authors used existing FEDformer (Zhou et al., 2022), DLinear (Zeng et al., 2023), or MICN (Wang et al., 2023) as representation networks to learn decomposed representation.  Given the serious concerned raised about the novelty, I recommend a reject.

**Justification For Why Not Higher Score:**

lack of novelty.

**Justification For Why Not Lower Score:**

N/A

---

### Decision · Program_Chairs · 2024-01-16

Reject